# The Relationship between Different Amounts of Physical Exercise, Internal Inhibition, and Drug Craving in Individuals with Substance-Use Disorders

**DOI:** 10.3390/ijerph182312436

**Published:** 2021-11-26

**Authors:** Tingran Zhang, Kun Wang, Ning Li, Chansol Hurr, Jiong Luo

**Affiliations:** 1Research Centre for Activity Detoxification, College of Physical Education, Southwest University, Chongqing 400715, China; zhangtingran1990@163.com (T.Z.); wangkun05282021@163.com (K.W.); 2Integrative Exercise Physiology Laboratory, Department of Physical Education, Jeonbuk National University, 567 Baekje-daero, Deokjin-gu, Jeonju-si 54896, Jeollabuk-do, Korea; li1990@jbnu.ac.kr

**Keywords:** SUD, amount of physical exercise, drug craving, internal inhibition, mediating effect

## Abstract

Purpose: To explore the relationship between different amounts of physical exercise and drug craving in individuals with substance-use disorders (SUD), and to reveal the mediating role of internal inhibition between physical activity and drug craving. Method: This study adopted the Physical Activity Rating Scale, Internal Inhibition Scale, and Drug Craving Scale to assess 438 cases of SUD in a compulsory isolation detoxification center in southwest China. Results: (1) The amount of physical exercise individuals with SUD engaged in was positively correlated with internal inhibition and negatively correlated with drug craving, while the amount of physical exercise was negatively correlated with drug craving. (2) The amount of physical exercise was able to negatively predict drug craving in addicts, the amount of physical exercise and internal inhibition were able to jointly predict drug craving, and internal inhibition played a mediating role between the amount of physical exercise and drug craving (the mediating effect was 0.22). (3) There was a dose-effect relationship regarding different amounts of physical exercises and drug craving. Internal inhibition did not mediate between a low amount of physical exercise and drug craving, it played a partial mediating role between a moderate amount of physical exercise and drug craving (the mediating effect was −0.19), and it played a partial mediating role between a high amount of physical exercise and drug craving (the mediating effect was −0.15). Conclusions: Physical activity has a positive effect on reducing drug craving in individuals with SUD. Moreover, in the process of sports rehabilitation for SUD, medium or high amounts of physical activity were required in order to effectively reduce and alleviate drug cravings.

## 1. Introduction

Drug abuse is a global problem that poses a threat to public health and social development. Drug addiction not only seriously damages the physical and mental health of individuals with substance-use disorders (SUD) but also causes various terrible social problems that seriously affect the social order [1]. According to the data, there were 2.404 million cases of SUD in China by the end of 2018, which accounts for 0.18% of the national population [2]. It should be noted that, in addition to the large total number of drug users, the high relapse rate is another major reason for the drug epidemic. At present, the international average drug relapse rate is nearly 91%, with the relapse rate remaining above 80% in countries with advanced drug rehabilitation technology, such as the United States, Germany, and Singapore. The drug relapse rate in China is around 90% [3]. Studies show that drug addiction and relapse behavior in cases of SUD are affected by multiple factors. Of these, drug craving is one of the most important factors concerning recurrent relapse [4]. Drug craving is defined as a strong and unquenchable desire for drug users to re-obtain the psychoactive substances they have experienced and the driving force to subconsciously pay excessive attention to drug-related cues and continue to use addictive drugs regardless of the consequences [5,6]. According to self-discrepancy theory, when an individual is craving drugs, a crack will appear between his/her real self and his/her supposed self, which makes the individual prone to generate negative emotions, such as anxiety and depression [7]. In addition, the more intense the drug craving, the higher the level of anxiety and depression [8]. Currently, the most commonly used rehabilitation treatment methods mainly include drug substitution therapy, comprehensive therapy, and psycho-behavioral therapy [9,10,11].

In recent years, exercise in medicine has attracted much interest. In medical research and clinical practice, it was proven that the rehabilitation effect of regular exercise could invigorate health, prevent illness, promote physical and mental health, and improve life quality [12,13]. In the field of drug addiction, exercise, a green and environmentally friendly rehabilitation method, has gradually gained recognition for its effectiveness in improving drug addiction. Studies demonstrated that exercise can accelerate the synthesis of dopamine by promoting the expression of tyrosine hydroxylase and stimulating the expression of the dopamine receptor-coupling protein, thus effectively slowing down various withdrawal symptoms caused by the sharp decrease in dopamine in the body at the early stage of drug withdrawal [14]. Among people addicted to smoking, regular aerobic exercise can effectively prevent or alleviate the negative emotions of the addicts and reduce their cravings for cigarettes [15]. Studies showed that after active aerobic exercise intervention, drug cravings in marijuana- and opioid-withdrawal patients are significantly reduced, and their withdrawal symptoms are significantly improved [16,17]. Zhao et al. [18] reported in a review study that exercise can effectively inhibit an individual with SUD’s psychological craving for drugs by regulating neurotransmitters and hormones in the brain. Ouyang et al. [19] demonstrated through the structural equation model that physical activity was significantly negatively correlated with drug cravings in Acquired Immune Deficiency Syndrome (AIDS) drug addicts, which can promote a reduction of drug cravings. Accordingly, this study proposes:

**Hypothesis** **1** **(H1).**
*Physical exercise has a negative predictive effect on drug craving in individuals with SUD.*


It is well known that long-term drug use seriously damages the user’s cognitive executive function, leading to abnormal activation of the prefrontal cortex, resulting in cognitive and behavioral abnormalities. As an important part of the executive function of the brain, internal inhibition is not only closely related to the prevention and treatment of attention deficit hyperactivity disorder (ADHD), substance abuse, and schizophrenia [20,21,22], but it has also been shown to be closely related to drug cravings [23]. In fact, after repeated drug use, drug abusers’ internal inhibition gradually declined under the influence of drugs, and when they were again exposed to the drug setting, it was difficult to suppress craving for drugs and the corresponding impulse to use drugs [24,25]. Moreover, the “incentive-sensitization” of the brain system is also considered to be the key to compulsive drug-seeking behaviors in addicts [26]. These are undoubtedly important reasons for the high relapse rates observed. In similar studies, it was found that addicted individuals with low self-control levels were more likely to have drug cravings and corresponding abuse behaviors, while those with high self-control levels showed fewer substance abuse behaviors [27]. It can be seen that the effective regulation of the internal inhibition of SUD is the key to reducing drug craving and relapse rates. Therefore, this study proposes:

**Hypothesis** **2** **(H2).**
*The internal inhibition of SUD has a negative predictive effect on drug craving.*


Interestingly, physical activity has been shown to correlate strongly with internal inhibition. Hillman et al. [28] reported that active participation in aerobic exercise can, to a certain extent, repair the impaired cognitive control ability of drug abusers and enhance the ability to inhibit the impulse to use drugs, thus achieving the goal of slowing down withdrawal symptoms. Roessler [16] conducted aerobic exercise intervention for 6 months (three times per week and 2 h per time) for drug abusers, such as those using marijuana and opioids, and the results showed that the self-control of drug patients was significantly enhanced and drug craving was significantly reduced. Wang et al. [29] believed that in a methamphetamine-dependent population, the enhancement of inhibition caused by acute exercise was a possible mechanism involved in the relationship between acute exercise and drug craving. The results suggest that the abilities of inhibition or self-control may have a mediating role in physical exercise and drug craving in drug abusers. Moreover, another question is to what extent physical activity has the best effect on internal inhibitions and drug cravings in individuals with SUD. Among college students who were addicted to smoking, Zhu et al. [30] found that, on the one hand, physical exercise could effectively reduce their craving to smoke by improving their self-control ability. On the other hand, there was a linear relationship between the amount of physical exercise, self-control, and smoking dependence, i.e., with the increase in the amount of physical exercise, the stronger the self-control ability of college students and the lower the dependence on smoking. Rong et al. [31] used functional near-infrared spectroscopy (fNIRs) technology to perform Stroop tasks on methamphetamine addicts and found that both moderate- and high-intensity aerobic exercises could increase the prefrontal cortex (PFC) activity of addicts, but moderate-intensity effects were mainly concentrated in the right dorsolateral prefrontal cortex (DLPFC), while high-intensity exercise also enhanced the nerve activation of the left DLPFC and the right ventrolateral prefrontal cortex (VLPFC). Another study found that both medium- and high-intensity long-term aerobic exercises could enhance the self-regulation function of addicts experiencing drug cravings by enhancing the alpha wave under conditions of drug-cue exposure [32]. Therefore, it is speculated that physical exercise, i.e., the amount of exercise or the exercise intensity, may have a “dose effect” in terms of reducing drug cravings in individuals with SUD. There are few studies that confirm whether, with the change in the amount of physical exercise, the addicts’ internal inhibition and drug craving change correspondingly. Meanwhile, while the positive effects of physical activity on reducing drug craving in individuals with SUD are widely recognized, few studies explore the key role of internal inhibition. Therefore, given the limitations of previous studies, this study adopted the structural equation path model to discuss the relationship between physical activity, internal inhibition, and drug craving and proposed:

**Hypothesis** **3** **(H3).**
*Physical exercise has a positive predictive effect on internal inhibition of SUD.*


**Hypothesis** **4** **(H4).**
*Internal inhibition plays a mediating role between the amount of physical exercise and drug craving, and the mediating role in medium and high amounts of exercise is stronger than in low amounts of exercise.*


## 2. Respondents and Methods

### 2.1. Respondents

This study adopted the cluster sampling method. The survey group was made up of individuals with SUD in the rehabilitation and consolidation area of the male and female compulsory isolation rehabilitation centers in Chongqing, China. Sampling surveys were conducted with brigade units (such as brigade 1, brigade 2, brigade 3, etc.). To ensure the validity and reliability of the questionnaires, the supervision personnel in the drug control center distributed the questionnaires on our behalf, and the dormitory units filled in and collected the questionnaires uniformly. Before filling out the questionnaires, all SUD signed the informed consent form. Five-hundred questionnaires were issued, and 482 were collected, with a collecting rate of 96.40%. After eliminating 44 invalid questionnaires that included unknown key information, we were left with 438 valid questionnaires, with an effective rate of 90.87%. Moreover, 176 were male, and 262 were female; the age was 34.72 ± 9.52 years old, the height was 1.61 ± 0.06 m, the weight was 58.67 ± 8.41 kg, and the drugs used were divided into 3 types (162 people reported using new drugs, such as methamphetamine, ecstasy; 157 people reported using traditional drugs, such as heroin, morphine; and 119 people reported using a mix: both new drugs and traditional drugs). 

### 2.2. Measurement Method

#### 2.2.1. Physical Activity Rating Scale (PARS-3)

The revised three-question test method was compiled by Liang [33]. It adopts the exercise intensity (for example, what do you think of the intensity of physical exercise you participate in), exercise duration (how long have you participated in physical exercise), and exercise frequency (how often do you participate in physical exercise every week) to observe exercise volume. Likert’s 5-point scale was adopted to measure the participation level of physical exercise from 1 to 5 points. Physical exercise score = exercise intensity score × (exercise duration score −1) × exercise frequency score, and the score range is 0–100 points. The grade of physical exercise is divided into a low amount of physical exercise, ≤19 points; medium amount of physical exercise, 20–42 points; and high amount of physical exercise, ≥43 points. The pre-test of the questionnaire demonstrated a high retest reliability and a correlation coefficient r = 0.82.

#### 2.2.2. Internal Inhibition Scale

This scale was mainly revised and compiled with reference to the Internal Inhibition Scale (Appendix A) of Jiang et al. [34]. This scale contains 25 items, such as *I don’t think I can easily behave well*, *I don’t think I can concentrate on anything easily*, etc., and it is quantified with a Likert 5-point measuring scale. The options *very much agree, relative agreement, general agreement, do not agree very much,* and *disagree* were scored 1–5, respectively. After the internal consistency test, 8 items were eliminated because of their small contribution. Therefore, the total score of internal inhibition was made up of the addition of 17 items, which constituted 2 dimensions. The internal consistency test results were as follows: the Cronbach’s α coefficient of self-control was 0.82 (6 items), and the deliberative Cronbach’s α coefficient was 0.91 (11 items), which was slightly higher than 0.89 of Jiang et al. [34]. The total score of internal inhibition was made up of the addition of 17 items (the scoring range was 17–85 points); the higher the score, the stronger the internal inhibition. The overall Cronbach’s α coefficient of internal inhibition was 0.91. Verification results of the measurement model were as follows: x^2^/df = 2.72, RMSEA = 0.04, AGFI = 0.96, CFI = 0.97, TLI = 0.98, IFI = 0.94, GFI = 0.97. This indicates that the questionnaire has good measurement validity and reliability. 

#### 2.2.3. Drug Craving Scale

This scale was mainly revised and compiled with reference to the Craving Beliefs Questionnaire (Appendix A) of Jiang et al. [34]. This scale contains 25 items, such as *taking drugs can make people temporarily forget their worries*, *taking drugs will fill me with excitement,* etc. The Likert 5-point measuring scale was adopted for quantification according to the choice *disagree, do not agree very much, general agreement, relative agreement, very much agree*, which were scored 1–5 points, respectively. After the internal consistency test, 3 items were eliminated because of their small contribution, so there were 22 valid items left, which constituted 3 dimensions. The internal consistency test results were as follows: Cronbach’s α coefficient of drug craving was 0.93 (5 items), Cronbach’s α coefficient of irrational belief was 0.93 (7 items), and Cronbach’s α coefficient of medication cognition was 0.96 (10 items). The total drug craving score was made up of the addition of 22 items (the scoring range was 21 to 110 points). The higher the score, the stronger the drug craving. The overall Cronbach’s α coefficient of the total score was 0.96. Verification results of measurement model were as follows: x^2^/df = 1.78, RMSEA = 0.04, AGFI = 0.98, CFI = 0.98, TLI = 0.95, IFI = 0.96, GFI = 0.98. This indicates that the questionnaire has good measurement validity and reliability.

### 2.3. Statistical Method

This study adopted SSPS 21.0 (IBM Corp.: Armonk, NY, USA) to conduct the statistical analysis of the data. Independent sample *t*-test and single-factor analysis were adopted to conduct the demographical variance analysis of internal inhibition and drug craving of SUD. Pearson correlation analysis was adopted to investigate the correlation between amount of exercise, internal inhibition, and drug craving. Amos 21.0 and Process were adopted to build a structural equation model and discuss the mediating effects. When the research hypothesis was tested, it was mainly based on the latest mediating effect testing process proposed by Wen et al. [35]. The significance level of all indicators was set at α = 0.05.

## 3. Result

### 3.1. Common Method Variance Test

All data in this study were collected by questionnaire self-reported by respondents. There may have been the risk of common method variance. To improve the rigor of this study, according to the research of the predecessors, Harman single-factor test [36] was adopted to conduct common method variance. The results show that there were 10 factors with an eigenvalue greater than 1, and the variance explained by the first factor was 29.93%, less than the critical standard of 40%, which proves that the common method has no significant variation.

### 3.2. Analysis of Drug Craving for SUD

The results of this study showed that, on the Likert scale of 5, the mean score of individuals with SUD’s drug craving was *M* = 41.74, indicating that, on the whole, drug craving of individuals with SUD in the rehabilitation consolidation period was at a moderate level, which means that physical detoxification significantly reduced addicts’ drug craving. To further explore the status of drug craving in addicts, this study also examined differences in gender and the type of drug. The results showed that the main effect of gender was not significant: male *M* = 40.898 and female *M* = 42.305 (*F* = 0.723, *p* = 0.396). The main effect of drug abuse type was not significant: *M*= 44.025 for new drugs, *M* = 39.312 for traditional drugs, and *M* = 41.832 for mixed drugs (*F* = 2.553, *p* = 0.079). The interaction effect between gender and drug abuse type was not significant (*F* = 0.345, *p* = 0.709).

### 3.3. Variance Analysis of the Influence of Different Amounts of Physical Exercise on Internal Inhibition and Drug Craving

Because the amount of physical exercise is an integral variable, to further discuss the difference in the influence of physical exercise on internal inhibition and drug craving for individuals with SUD, according to the research of the predecessors, according to the score, this study divided the amount of physical exercise into three levels [33] as follows: a low amount of exercise (240 people), a medium amount of exercise (99 people), and a high amount of exercise (99 people).

The results of the single-factor analysis (Table 1) showed that individuals with SUD with different amounts of physical exercise showed significant differences in internal inhibition and drug craving. Firstly, in terms of internal inhibition, there were significant differences between a low amount of exercise and a medium amount of exercise, and a low amount of exercise and a high amount of exercise in addicts (*F* = 50.983, *p* < 0.001, low amount of exercise < medium amount of exercise, low amount of exercise < high amount of exercise), while there were no significant differences between a medium amount of exercise and a high amount of exercise (*p* > 0.05). Secondly, in terms of drug cravings, there were significant differences between a low amount of exercise and a medium amount of exercise and a high amount of exercise in addicts, respectively. (*F* = 12.760, *p* < 0.001, low amount of exercise > medium amount of exercise, low amount of exercise > high amount of exercise), while there were no significant differences between a medium amount of exercise and a high amount of exercise (*p* > 0.05).

### 3.4. Correlation Analysis of Physical Exercise, Internal Inhibition, and Drug Craving

The correlation analysis showed (Table 2) that there was a significant correlation between the amount of physical exercise, internal inhibition, and drug craving. Among them, exercise amount was positively correlated with internal inhibition (*r* = 0.444) and negatively correlated with drug craving (*r* = −0.220); drug craving was negatively correlated with internal inhibition (*r* = −0.391). The correlated coefficient of the variables was significant, which provides a good basis for the subsequent testing of the mediating effect of internal inhibition between physical exercise and drug craving.

### 3.5. Test of Dual Mediating Effect of Internal Inhibition in Physical Exercise and Drug Craving

The regression equation was adopted to test the relationship between physical exercise and drug craving, and the mediating effect test method by Baron and Kenny [37] was adopted. This mainly includes three steps: first, independent variables have an impact on dependent variables, and the regression coefficient is significant. Second, independent variables have an impact on mediating variables, and the regression coefficient is significant. Third, the joint influence of independent variables and intermediary variables on dependent variables is significant. The influence of the intermediary variable on the dependent variable must reach a significant level, i.e., if the influence of the independent variable on the dependent variable is significant, the intermediary variable plays a completely intermediary role; if the influence of the independent variable on the dependent variable is reduced, the intermediary variable plays an intermediary role. The regression analysis results of this study showed the following (Table 3): 

Equation (1): When the amount of physical exercise was used as the predictive variable and drug craving was used as the dependent variable, the amount of physical exercise had a significant negative predictive effect on drug craving (*β*1 = −0.220, *t* = −4.719, *p* = 0.000).

Equation (2): When the amount of physical exercise was used as the predictive variable and the internal inhibition was used as the dependent variable, the amount of physical exercise had a significant positive predictive effect on the internal inhibition (*β* = 0.444, *t* = 10.349, *p* = 0.000).

Equation (3): When the internal inhibition and the amount of physical exercise were used as prediction variables at the same time, and drug craving was used as the dependent variable, drug craving could be significantly predicted, and the internal inhibition had a significant negative predictive effect on drug craving (*β* = −0.365, *t*= −7.419, *p* = 0.000), but the predictive effect of the amount of physical exercise on drug craving turned from significant to insignificant (*β*2 = −0.058, *t* = −1.188, *p* = 0.236), indicating that internal inhibition plays a complete mediating role between physical exercise and drug craving.

Figure 1 shows the structural equation model used to test the mediating effect of internal inhibition between physical exercise and drug craving. The fitting indexes of the structural equation model were x^2^/df = 1.019, RMSEA = 0.007, GFI = 0.995, TLI = 1.000, NFI = 0.992, and AGFI = 0.984, respectively. The model had a good fitting degree, indicating that it was suitable to test the mediating effect. The result showed that the direct pathway coefficients of the amount of physical exercise on drug craving were significant (*β*1 = −0.236, *SE* = 0.016, *p* < 0.001), the pathway coefficient of the amount of physical exercise and internal inhibition was also significant (*β*= 0.490, *SE* = 0.009, *p* < 0.001), and the pathway coefficient of internal inhibition and drug craving was significant (*β*= −0.449, *SE* = 0.162, *p* < 0.001), but the pathway coefficient of the amount of physical exercise and drug craving changed from significant to insignificant (*β*2 = −0.022, *SE* = 0.019, *p* > 0.05), indicating that internal inhibition plays a complete mediating role between physical exercise and drug craving; the mediating effect was 0.490 × 0.449 = 0.22. Therefore, Hypotheses H1, H2, and H3 in this study were confirmed.

### 3.6. Test of the Dual Mediating Effect of Internal Inhibition in Different Amounts of Physical Exercise and Drug Craving

As we have shown above, internal inhibition has a completely mediating effect between the amount of physical exercise and drug craving. However, previous studies showed that the amount of physical exercise is a complex variable, and there is often a dose effect regarding amount of physical exercise. Given this, to further clarify the impact of different amounts of physical exercises on drug craving and to explore the most suitable exercise prescription for SUD exercise rehabilitation, in this study, Model 4 of SPSS macro compiled by Hayes [38] was adopted to estimate the 95% confidence interval of the mediating effect and to test the mediating effect using 5000 samples. If the 95% confidence interval of the mediating effect does not include 0, the mediating effect is significant and vice versa. In addition, as suggested by Hayes, the mediating effect test in this study was conducted under the condition of controlling for statistical variables, such as gender, age, and drug-use type.

The results of the regression analysis showed the following (Table 4): (1) a low amount of physical exercise cannot effectively predict the internal inhibition (*β* = 0.213, *p* > 0.05); internal inhibition could effectively negatively predict drug craving (*β* = −0.507, *p* < 0.001); a low amount of exercise and internal inhibition predict drug craving at the same time; and a low amount of exercise cannot effectively predict drug craving (*β* = −0.264, *p* > 0.05). (2) A medium amount of exercise can positively predict internal inhibition (*β* = 0.483, *p* < 0.05); internal inhibition could effectively negatively predict drug craving (*β* = −0.389, *p* < 0.001); the amount of physical exercise and internal inhibition predict drug craving at the same time; and a medium amount of exercise can effectively negatively predict drug craving (*β* = −0.412, *p* < 0.05). (3) A high amount of exercise can effectively predict internal inhibition (*β* = 0.354, *p* < 0.001); internal inhibition can negatively predict drug craving (*β* = −0.417, *p* < 0.001); a high amount of exercise and internal inhibition predict drug craving at the same time; and a high amount of exercise can effectively negatively predict drug craving (*β* = −0.178, *p* < 0.05).

Figure 2, Figure 3 and Figure 4 show the acting pathway and corresponding pathway coefficient values of the low, medium, and high amounts of exercise on drug craving. If the bootstrap 95% interval of indirect effect caused by internal inhibition force does not contain 0, this variable has a mediating effect. In this study, there were two effective mediating effect models: (1) The low amount of exercise → internal inhibition → drug craving; the confidence interval of this pathway contains 0, indicating that the mediating effect of internal inhibition between low exercise volume and drug craving is not significant. (2) The medium amount of exercise → internal inhibition → drug craving; the confidence interval of this pathway does not contain 0, indicating that the mediating effect of internal inhibition between medium exercise volume and drug craving is significant (standardized effect value: −0.389 × 0.483 = −0.19, accounting for 32% of the total effect). A medium amount of exercise has a significant direct predictive effect on drug craving; therefore, internal inhibition has a partial mediating effect between medium exercise volume and drug craving. (3) The high amount of exercise →internal inhibition →drug craving; the confidence interval of this pathway does not contain 0, indicating that the mediating effect of internal inhibition between low exercise volume and drug craving is significant (standardized effect value: −0.417 × 0.354 = −0.15, accounting for 45% of the total effect). A high amount of exercise has a significant direct predictive effect on drug craving; therefore, internal inhibition has a partial mediating effect between high exercise volume and drug craving. Therefore, hypothesis H4 in this study was confirmed.

## 4. Discussion

### 4.1. Direct Impact of Physical Exercise on Drug Craving

This study found that physical exercise is significantly negatively correlated with drug craving for drugs addicts, which proves that physical exercise has a positive impact on drug craving for individuals with SUD, which is consistent with previous research [18,39,40]. Roberts et al. [41] reported that more than 80% of studies show that physical exercise can effectively relieve drug cravings for individuals with SUD. Moreover, previous studies found that physical exercise has a positive impact on drug craving for addicts using different drug types, and there is no difference in the influence of drug type. For example, Linke et al. [15] found that, among smokers who are addicted to smoking, regular participation in physical exercise can effectively prevent or reduce the negative emotions when quitting smoking and, at the same time, reduce the desire for cigarettes. Roessler [16] found that after active aerobic exercise, drug cravings in marijuana and opioid addicts were significantly reduced, and their withdrawal symptoms were significantly improved. Wang et al. [42] found that, after three months of cycling, jogging, or jumping rope exercise intervention, the drug craving of methamphetamine addicts exhibited a significant downward trend. In addition, Kulig et al. [43] reported that for physical activity, there were no significant gender differences in reducing the desire for in the individual with SUD. This study conducted comprehensive research on the relationship between physical activity levels, participating status in sports teams, and drug use, the results found that, among young males and females, those who reported higher physical activity and higher participation in sports teams showed lower levels of drug craving and drug-use rate. It should be noted that we found that physical exercise was also significantly negatively correlated with the three dimensions of drug cravings (drug cognition, irrational beliefs, craving level), which is consistent with the research of Ouyang et al. [19]. We speculate that addicts’ craving for drugs is affected by many factors, and physical exercise can, to a certain extent, change drug addict cognition towards drugs, correct misconceptions, and thus reduce cravings.

It is worth noting that both individual and collective physical exercise activities contain elements, such as exercise intensity, exercise frequency, and exercise time, which involve whole-state and continuous processes, i.e., the amount of physical exercise [44]. Adhering to regular physical exercise is to carry out an organic combination of all physical exercise elements to reach the maximum benefits of physical exercise. The results of this study showed that individuals with SUD adhering to different amounts of physical exercise had significant differences in terms of drug craving, and addicts engaged in low amounts of exercise had significantly higher levels of cravings than addicts involved in medium or high amounts of exercise. A further analysis showed that low amounts of exercise had no significant effect on drug craving, while medium and high amounts of exercise could directly negatively predict drug craving. That is to say, the higher the amount of exercise in which one is engaged, the lower the drug craving. Zhu et al. [30] reported similar findings in college students addicted to smoking, showing that the amount of physical exercise is negatively correlated with their dependence on smoking. That is to say, the higher the amount of physical exercise, the weaker the smoking dependence of college students. To further explore the influence of different exercise volumes/intensities on the drug addiction formation process, Peterson et al. [45,46] found that free-wheeling exercise for a longer time (6 h/day) was more effective in reducing the cocaine-seeking behavior of rats. Therefore, physical exercise can be an effective means of rehabilitation in terms of reducing drug cravings for individuals with SUD, and medium and high amounts of exercise were more effective in rehabilitation. The results suggest that the necessary amount of exercise should be maintained in future interventions in detoxification scenarios to ensure rehabilitation benefits.

### 4.2. Mediating Effect of Internal Inhibition on the Drug Craving Pathway Affected by Physical Exercise

In this study, the internal inhibition force was used as a mediating variable in the structural equation model to explore the relationship between the amount of physical exercise, internal inhibition, and drug craving. The results showed that internal inhibition has a complete mediating effect between the amount of physical exercise and drug craving, i.e., with the increase in the amount of physical exercise, the addicts’ internal inhibition improved, and drug craving was effectively reduced. This suggests that internal inhibition may be an important variable affecting drug cravings in individuals with SUD. In fact, among normal people, there are a large number of studies that found that physical exercise could bring about physical changes in internal inhibition-related brain regions, promoting inhibitory control function [47,48,49]. Among smoking-addicted individuals, the self-control resource model showed that adhering to physical exercise can improve an individual’s self-control resource [50,51], while in most cases, people with high self-control have low cigarette dependency, and self-control has a mediating effect on physical exercise and cigarette dependency [30]. Among individuals with SUD, appropriate physical exercise could improve inhibitory and other cognitive brain areas by activating the anterior cingulate, thus inhibiting drug craving [52]. Research shows that active aerobic exercise produced effective stimuli to the cranial nerve of meth addicts, making them perform better in inhibition tasks. It also effectively promoted and improved the corresponding functional brain regions and inhibitory ability and significantly reduced drug craving in meth addicts, thus reducing the relapse tendency [24,29,53]. However, it should be noted that there may be differences in the internal inhibition power of different groups. Therefore, the items on the scale should be appropriately revised in order to make them more targeted during testing. In general, people with stronger executive control are more likely to take part in physical exercises as a result of self-discipline; they will, in turn, have better control over their behavior and drug cravings. This could directly explain the linear relationship between physical exercise, internal inhibition, and drug craving, thus revealing the important role of internal inhibition on the drug craving pathway as affected by physical exercise.

To further explore the relationship between physical exercise and drug craving for individuals with SUD and to confirm whether the mediating effect of internal inhibition on physical exercise and drug craving differs according to changes in the amount of exercise, this study divided the amount of physical exercise into low, medium, and high levels. The results demonstrate that internal inhibition has no mediating effect on low amounts of exercise and drug craving, and the direct effect of low amounts of exercise on drug craving is not significant. Moreover, internal inhibition has a partial mediating effect on medium and high amounts of exercise and drug craving, indicating that low amounts of exercise have no significant impact on drug craving, while medium and high amounts of exercise could positively affect the inhibition effect. Previous studies show that physical exercise for 30~40 min at least three times a week can produce a good inhibition effect [54]. In addition, increasing the volume or duration of exercise may improve the effectiveness of physical exercise in cases of drug addiction, i.e., the intervention effect of exercise may be the dose effect [45,55]. Because the amount of physical exercise is a comprehensive variable made up of exercise duration, exercise frequency, and exercise intensity, at present, most researchers adjust the intensity to explore the dose effect of the amount of physical exercise, thus controlling the exercise duration and frequency under a consistent premise to discuss the effect of internal inhibition on exercise intensity and drug craving in meth addicts. Wang et al. [53] divided exercise intensity into low, medium, and high intensity, and the results showed that medium- and high-exercise groups had better control over drug craving, and their drug craving was significantly higher than those in the low exercise group. Daniel et al. [56] explored a three times per week, four-week, high-intensity physical exercise regime with alcohol and cocaine addicts. The results demonstrate that, after high-intensity exercise, the addicts exhibited more activity in the prefrontal cortex during cognitive tests, their performance in cognitive tests was enhanced and prefrontal cortex oxygenated hemoglobin was increased, and anxiety levels were significantly reduced, which may be an effective way to reduce drug craving. In summary, in addition to low amounts of exercise, internal inhibition in relation to medium and high amounts of exercise demonstrated a mediating effect. Moreover, there was a dose effect between the amount of physical exercise, internal inhibition, and drug craving and medium and high amounts of exercise had a significant effect on the reducing drug craving in individuals with SUD. This indicates that the individual with SUD should engage in the appropriate amount of exercise in order to benefit fully from drug rehabilitation.

It should be noted that this study was a cross-sectional design study that used questionnaires to investigate physical exercise and the relations between internal inhibition and drug craving. Future research should adopt a more longitudinal design with more quantitative data in order to further elucidate the results herein.

## 5. Conclusions and Limitations

### 5.1. Conclusions

(1)The amount of physical exercise was positively correlated with internal inhibition of the individual with SUD and negatively correlated with drug craving. Internal inhibition was negatively correlated with drug craving;(2)Internal inhibition demonstrated a complete mediating effect between the amount of physical exercise and drug craving. Physical exercise can influence drug craving in the individual with SUD through the mediating variable of internal inhibition;(3)There is a dose effect related to the effect of different amounts of physical exercise on drug craving. Internal inhibition did not play a mediating role between low amounts of exercise and drug craving, but it played a partial mediating role between a medium amount of exercise and drug craving and played a partial mediating role between high exercise volume and drug craving. Therefore, medium and high amounts of physical exercise are effective ways for individuals with SUD to improve their internal inhibition and thus reduce and alleviate their drug cravings.

### 5.2. Limitations and Further Research Directions

(1)As this study was a horizontal study, the results were subjective, and a deeper causal relationship could not be ascertained. Future research should be longitudinal and empirical in design and scientifically designed exercise prescriptions based on the results of this research could be utilized to better reveal the causal relationships between variables. In this way, the relationship between physical exercise, intrinsic inhibition, and drug craving can be objectively explored.(2)This study focused on the mediating effect of internal inhibition between different amounts of exercise and drug cravings without adding other mediating variables. This has certain limitations. More mediating or regulating variables can be explored in the future.

## Figures and Tables

**Figure 1 ijerph-18-12436-f001:**
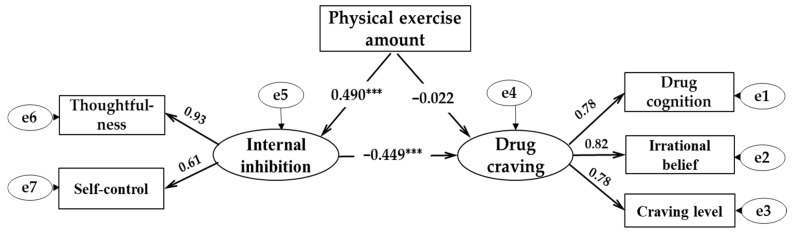
The structural equation model of the effect of physical activity on internal inhibition and drug craving. *** *p* < 0.001.

**Figure 2 ijerph-18-12436-f002:**
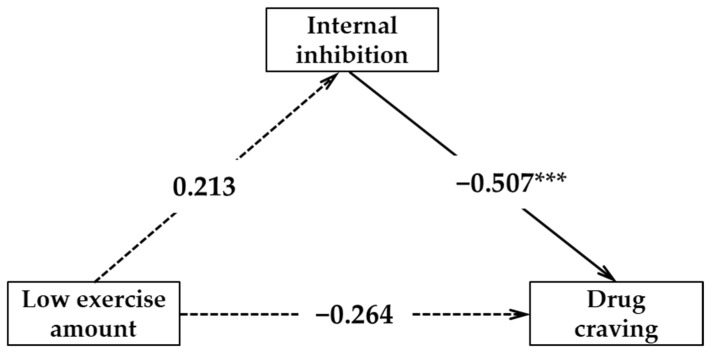
The mediating model of internal inhibition between lower physical exercise amount and drug craving. *** *p* < 0.001.

**Figure 3 ijerph-18-12436-f003:**
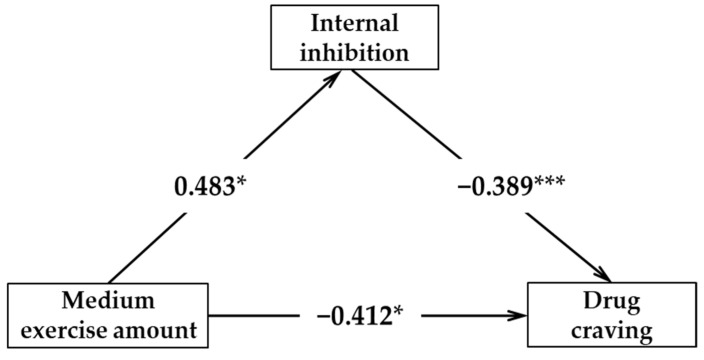
The mediating model of internal inhibition between medium physical exercise amount and drug craving. * *p* < 0.05, *** *p* < 0.001.

**Figure 4 ijerph-18-12436-f004:**
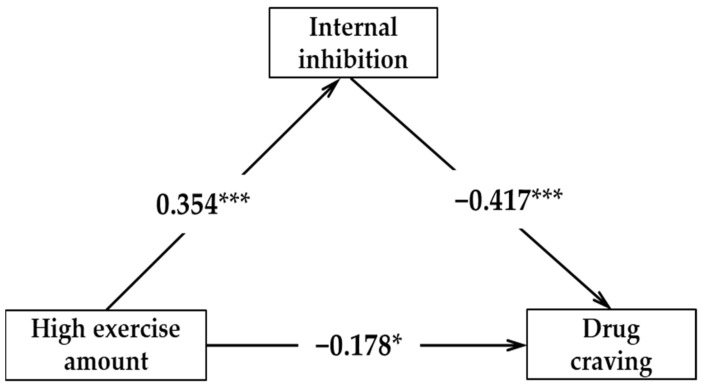
The mediating model of internal inhibition between high physical exercise amount and drug craving. * *p* < 0.05, *** *p* < 0.001.

**Table 1 ijerph-18-12436-t001:** Variance analysis of the influence of the amount of physical exercise on internal inhibition and drug craving (n = 438).

Variable	Amount of Physical Exercise	M	SD	F	LSD Multiple Comparisons
Internal inhibition	(a) low amount of exercise	46.638	10.745	50.983 ***	b > a, c > a
	(b) medium amount of exercise	58.727	15.546		
	(c) high amount of exercise	58.818	13.126		
Drug craving	(A) low amount of exercise	45.521	20.567	12.760 ***	A > B, A > C
	(B) medium amount of exercise	35.030	15.473		
	(C) high amount of exercise	39.283	14.313		

*** *p* < 0.001.

**Table 2 ijerph-18-12436-t002:** Correlation between physical exercise, internal inhibition, and drug craving (n = 438).

Variable	M	SD	1	2	3	4	5	6	7	8
1. Exercise amount	24.90	24.19	1							
2. Internal inhibition	52.12	13.88	0.444 ***	1						
3. Drug craving	41.74	18.72	−0.220 ***	−0.391 ***	1					
4. Self-control	17.29	5.26	0.277 ***	0.800 ***	−0.278 ***	1				
5. Thoughtfulness	34.84	10.17	0.462 ***	0.951 ***	−0.389 ***	0.574 ***	1			
6. Drug cognition	18.00	9.60	−0.187 ***	−0.313 ***	0.907 ***	−0.206 ***	−0.321 ***	1		
7. Irrational belief	15.20	7.23	−0.197 ***	−0.375 ***	0.852 ***	−0.275 ***	−0.369 ***	0.602 ***	1	
8. Craving level	8.54	4.66	−0.194 ***	−0.344 ***	0.827 ***	−0.265 ***	−0.332 ***	0.651 ***	0.634 ***	1

*** *p* < 0.001. “Thoughtfulness” and “self-control” are the two sub-dimensions of “internal inhibition”; “drug cognition”, “irrational belief”, and “craving level” are the three sub-dimensions of “drug craving”.

**Table 3 ijerph-18-12436-t003:** Test equation of exercise amount, internal inhibition, and drug craving.

Equation	Dependent Variable	Independent Variable	β	t	p	R	R^2^	F	p	95% CI
Equation (1)	Drug craving	Amount of exercise	−0.220	−4.719	0.000	0.220	0.049	22.266	0.000	(−0.228, −0.107)
Equation (2)	Internal inhibition	Amount of exercise	0.444	10.349	0.000	0.444	0.197	107.093	0.000	(0.409, 0.500)
Equation (3)	Drug craving	Internal inhibition	−0.365	−7.419	0.000	0.394	0.155	40.034	0.000	(−0.652, −0.342)
		Amount of exercise	−0.058	−1.188	0.236					(−0.111, 0.019)

**Table 4 ijerph-18-12436-t004:** Regression analysis of the dual mediating effect of internal inhibition in different amounts of physical exercises and drug craving (N = 438).

Variable	Internal Inhibition			Drug Craving		
	t	β	95%CI	t	β	95%CI
Low amount of exercise Internal inhibition	1.904	0.213	(−0.007, 0.433)	−1.268 −4.220	−0.264 −0.507 ***	(−0.675, 0.146) (−0.743, −0.270)
R^2^	0.015			0.082		
F	3.625			10.522 ***		
Medium amount of exercise Internal inhibition	2.212	0.483 *	(0.050, 0.916)	−2.032 −4.223	−0.412 * −0.389 ***	(−0.814, −0.010) (−0.571, −0.206)
*R* ^2^	0.048			0.220		
*F*	4.893 *			13.529 ***		
High amount of exercise Internal inhibition	4.833	0.354 ***	(0.209, 0.499)	−2.074 −3.902	−0.178 * −0.417 ***	(−0.349, −0.008) (−0.629, −0.205)
*R* ^2^	0.194			0.256		
*F*	23.362 ***			16.539 ***		

Note: * *p* < 0.05, *** *p* < 0.001.

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
