# Peer review of "The Relationship between Different Amounts of Physical Exercise, Internal Inhibition, and Drug Craving in Individuals with Substance-Use Disorders"

_ijerph, 2021, doi:10.3390/ijerph182312436_

Round 1
Reviewer 1 Report
No comments for authors
Author Response
Thank you very much for your approval of our manuscript.
Reviewer 2 Report
The authors addressed minor comments within the manuscript. However, there remain major concerns which should be addressed. Otherwise I get the impression that the authors will not or are not able to address this issues.
It seemed that the authors used a questionnaire for internal inhibition which has been investigated for validity and reliability in a study by Jiang et al., 2000. However, the authors removed items. The author stated that this is in line with “general scientific research”. Deleting items from a questionnaire that has been investigated with regard to validity and reliability in prior studies is not in line with good scientific research.
- The authors should clearly state within the manuscript whether the applied questionnaire had been investigated before with regard to validity and reliability (in the study by Jiang et al., 2000).
- If so, the removal of items should be justified and discussed in manuscript.
- If not, it should be discussed that the validity and reliability of the internal inhibition questionnaire are questionable.
The construct validity of the craving questionnaire remains questionable. Craving is a symptom of an addiction (c.f. DSM-5 & ICD-11). Individuals with substance related disorders could crave for different drugs (e.g. alcohol, nicotine, methamphetamine, ecstasy, heroine, or morphine). It seems like new insight to the authors that alcohol is a drug. Regardless of the drug, craving always has the same symptoms. Sub-facets (=subscales) of craving are always described in the same way in addiction literature, e.g. Reward and relief and obsessive craving (Verheul et al., 1999). However, the authors state that irrational beliefs and medication cognitions are subscales of craving. This is not in line with craving literature and the statement that this is “more suitable for Chinese drug addicts” is not valid. However, the authors stated that the reliability and validity of the scale has been investigated in prior studies (https://doi.org/10.16099/j.sus.2020.10.006; https://doi.org/10.3389/fpsyg.2019.01928). However, in the cited studies no items were deleted in favor for internal consistency.
- Please provide a translation of all questionnaires in the appendix of the manuscript that the reader gets an impression of the validity.
- Please discuss in the manuscript that the craving subscales resemble more symptom severity of drug addiction than craving.
- justify within the manuscript why items had been deleted in favor for internal consistency (see above)
- Please state differences between the study by Wang et al., 2019 and the current study.
Please provide statistics for the indirect path within the mediations
English must be improved significantly. Please proofread the manuscript by an English expert.
Author Response
Thank you very much for your suggestions on our manuscript. We have carefully revised the manuscript according to your suggestions. Please refer to the attachment and manuscript for the revised content.

Round 2
Reviewer 2 Report
Thank you for addressing the comments
Author Response
1.Does the introduction provide sufficient background and include all relevant references?
Reply: Thank you for your suggestions. We have adjusted and supplemented the foreword accordingly. Please check the green font part of the manuscript.
2.Are the results clearly presented?
Reply: Thank you for your suggestion. We have partially reorganized and adjusted the results. Please check the green font part of the manuscript.

This manuscript is a resubmission of an earlier submission. The following is a list of the peer review reports and author responses from that submission.
Round 1
Reviewer 1 Report
congratulations to the authors for such an excelent manuscript.
Our research griup is actually looking into personality traits, hrv and drug abuse among young populations and this research was indeed interesting and of help.
Yet, my only concer to the authors would be to include a part of practical applications, of how this information may help in further interventions
Reviewer 2 Report
The manuscript explore the relationship between different physical exercise amount and drug craving of drug addicts. Some minor comments: The author should define the meaning of abbreviation when its first appearance (ADHD).
The purpose of the paper is to demonstrate that internal inhibition plays a mediating role between the physical exercise and drug craving. The study involved male and female detoxification addicts in rehabilitation detoxification center in a certain area of southwest China.
To conduct demographical variance analysis of internal inhibition and drug craving of drug addicts, were used independent sample T-test and single factor analysis as statistical methods.
At the end, the authors show a positive relationship between medium or high-level physical activity and drug craving of drug addicts.
Comments:
Although the correlation between the positive effects of physical activity on reducing drug craving in drug addicts is very discussed there are few articles on the key role of internal inhibition. The article is interesting and the limitations of this study are already considered by the authors.
Only small improvements are recommended as follows:
- Table 1 is not included in the paper, although it is mentioned in the text;
- Please explicit the acronym “ADHD” page 2 line 39;
- Please explicit the acronym “AIDS” page 3 line 16;
- Please explicit the acronyms “RMSEA”, “AGFI”, “CFI”, “TLI”, “IFI”, “GFI” page 4 line 21;
- The format of all references should follow the style request of the journal;
- The contribution of each author to the article should be specified;
- Minimal revision of English. Spelling of “activite” page 10 line 51;
- Insert space between Zhao et al. and (2018) page 2 line 30;
- Insert space between Wang et al. and (2016) page 10 line 46.
- The contribution of each author to the article was not specified
- The format of references should follow the style request of the journal.
Reviewer 3 Report
The authors investigate the relationship between different amount of physical exercise, internal inhibition and drug craving in a sample of individuals with different substance-use disorders during detoxification. The results show correlations between those variables and there was a mediating effect of internal inhibition on the relationship between exercise amount and craving.
The study is interesting, but there are some major concerns with regard to the construct validity and interpretation of the study:
- It seems that the authors use questionnaires for craving and internal inhibition, which were validated by Jiang et al. (2000). However, the authors eliminate items which did not show internal consistency in an exploratory factor analysis. This is quite unusual for prior validated questionnaires. Please comment.
- The construct validity of the craving scale is questionable. Subscales such as irrational beliefs, medication cognitions and drug craving are not reported as facets of the carving construct (https://doi.org/10.1111/j.1749-6632.2011.06298.x; https://doi.org/10.1093/alcalc/agg005). It seems that scales and items resemble an instrument for assessing symptom severity of drug dependence. Please comment
- The construct of internal inhibition seems to resemble general executive functioning. However, It could be argued that people with greater executive control are more likely to discipline themselves to exercise and have greater control over their behavior and of their drug symptom severity. That should also be discussed, beyond the assumption that exercise amount leads to higher executive function and thus provide lower symptoms of drug use.
- The mediating assumptions of internal inhibition should be stated more clearly. Right now also a moderating effect of internal inhibition on the relationship between exercise amount and symptom severity of drug use could be assumed (p.3, top: “But to what extent does physical activity have the best effect on the internal inhibitions and drug cravings of drug addicts?” or p. 3, middle: “With the change of physical exercise amount, there are few studies to confirm whether the drug addicts’ internal inhibition and drug craving will change correspondingly.). However, also the moderated regression should be reported and in case of interaction effects, simple slope analyses should be provided.
- The sampling procedure is completely incomprehensible. What kind of cluster sampling has been done? Are the cluster samples from one site or is it a multi-site trial? Please specify. Please provide more information for the type of drugs
- Table 1 is missing.
- Table 3 is incomprehensible. Exercise volume = Physical exercise score? Items 3 and 4 are subscales of 2 and 6,7,8 are subscales of 5? Please indicate in a more precise manner. otherwise the reader gets confused.
- Mediation: Please provide statistics for the indirect effect of the mediations.
- 1 -4: Would benefit from providing Predictior in the left corner, Mediator above and Dependent variable in the right corner.
- 5, middle: “Due to physical exercise amount is an integral variables”. Please provide more information about integral variable. Did you mean continuous variable or categorical? What level of measurement has it?
- 2, bottom: “In fact, after repeated use of drugs, drug abusers' internal inhibition will gradually decline under the influence of drugs. When they are exposed to the drug scene again, it is difficult to suppress the craving for drugs and the corresponding impulse to use drugs (Wang et al., 2015a)…” Please provide a more sophisticated theory for drug craving, symptoms of drug use and executive control, e.g. Incentive sensitization (Robbinson & Berridge, 1993) or the IRISA model (Volkow et al., 2010).
- There are many language issues, the manuscript should be proofread by an English expert, e.g.:
- 404 million drug addicts --> What kind of number is this? 2.5 million
- drug-related clues --> cues
- when an individual is craving for drugs
- healing drug craving --> craving is not a condition or disorder!
- drug addicts than nicotine or alcohol addicts --> also nicotine and alcohol are drugs
- drug addicts --> wording is stigmatizing, please provide more neutral terms such as individuals with substance use disorder (SUD) or something else…
- the stronger the internal inhibition people has --> have?
- All data in this study is collected --> are…
- Please provide statistic coefficients in italic
- higher the amount of physical exercise one has --> ?
- college students who depend on smoking --> odd language